# The Effects of Autonomic Dysfunction on Functional Outcomes in Patients with Acute Stroke

**DOI:** 10.3390/brainsci13121694

**Published:** 2023-12-08

**Authors:** Kyoung Hyeon Cha, Nae Yoon Kang, Sungchul Huh, Sung-Hwa Ko, Yong-Il Shin, Ji Hong Min

**Affiliations:** 1Department of Rehabilitation Medicine, Pusan National University Yangsan Hospital, 20 Geumo-ro, Mulgeum-eup, Yangsan 50612, Republic of Korea; dr.khcha@gmail.com (K.H.C.); dr.knybs12@gmail.com (N.Y.K.); dr.huhsc@gmail.com (S.H.); ijsh6679@gmail.com (S.-H.K.); rmshin01@gmail.com (Y.-I.S.); 2Research Institute for Convergence of Biomedical Science and Technology, Pusan National University Yangsan Hospital, 20 Geumo-ro, Mulgeum-eup, Yangsan 50612, Republic of Korea; 3Department of Rehabilitation Medicine, Pusan National University School of Medicine, 2 Busandaehak-ro 63beon-gil, Geumjeong-gu, Busan 46241, Republic of Korea

**Keywords:** autonomic nervous system diseases, stroke, treatment outcome, prognosis

## Abstract

Autonomic dysfunction is a common complication of acute stroke, which impairs functional outcomes and increases mortality. There is a lack of well-established knowledge regarding the influence of autonomic dysfunction in patients with acute stroke. This study aims to investigate the impact of the severity of autonomic dysfunction on functional outcomes in patients with acute stroke. A retrospective analysis was conducted at a single center, involving 22 patients diagnosed with acute stroke. The severity of autonomic dysfunction was evaluated based on the Composite Autonomic Scoring Scale (CASS). The modified Barthel Index, Berg Balance Scale, Functional Ambulatory Category, and modified Rankin Scale were designated as functional outcome measures. The impact of the severity of autonomic dysfunction on functional outcomes was analyzed using one-way analysis of covariance (ANCOVA). A statistically significant difference was observed between the initial and follow-up functional outcomes based on the severity of autonomic dysfunction. This study presents evidence that the severity of autonomic dysfunction influences functional prognosis in patients with acute stroke. The findings will serve as additional considerations for the rehabilitation of patients with acute stroke.

## 1. Introduction

The numerous nervous system injuries caused by stroke have a significant impact on a patient’s life. Stroke-related disabilities result in a decrease in a patient’s activities of daily living (ADL), balance, and ambulation. Prognosis prediction in patients with stroke may assist in setting recovery goals and guiding treatment strategies. Many studies have investigated factors to predict the prognosis after stroke onset and have aimed to predict clinical outcomes based on the timing and location of stroke occurrence, severity, initial National Institutes of Health Stroke Scale score (NIHSS), and the presence of autonomic dysfunction [1,2,3]. Stroke damages the central nervous system, leading to peripheral nervous system impairments, including autonomic nervous system dysfunction. Previous studies have highlighted the importance of autonomic dysfunction. In a prospective observational cohort study, autonomic dysfunction was highly prevalent in patients with mild acute ischemic stroke and persisted for 6 months. The prevalence of autonomic dysfunction in patients with mild acute stroke with a NIHSS score of 1–4 can range from 94 to 100%, depending on the type of autonomic function test [4,5].

Autonomic dysfunction is a common complication of acute stroke, which impairs functional outcomes, increases mortality [4], and occurs in approximately 25–76% of patients with acute stroke [5]. Autonomic dysfunction due to stroke affects respiratory, gastrointestinal, sudomotor, and urogenital functions through the dysregulation of the sympathetic and parasympathetic nervous systems [6,7]. Additionally, autonomic dysfunction affects the cardiovascular system by increasing the incidence of arrhythmias, myocardial infarction, and variability in blood pressure and heart rate [6]. Post-stroke, the incidence of cardiac arrhythmias ranges from 17 to 80% [6]. Furthermore, cardiovascular dysregulation can increase sympathetic tones and lead to unfavorable complications [7]. Sweating dysfunction is a common symptom of autonomic disturbance [7], and according to a prospective study, hyperhidrosis was observed in 55% of patients with acute brain infarction on the hemiplegic side. However, the pathogenesis of hyperhidrosis is still not fully understood. Although Labar et al. reported an association between hyperhidrosis and severe neurological deficits and poor prognosis, further research is required for clinical application [8,9]. Autonomic dysfunction affecting the gastrointestinal system often results in gastroparesis, which manifests as constipation, vomiting, and abdominal distension. Additionally, autonomic dysfunction in the gastrointestinal system may cause dysphagia, which is associated with poor nutrition intake. Urinary incontinence, retention, and other voiding deficits are also commonly observed complications [6]. Despite the potential severity of symptoms resulting from autonomic dysfunction, they are frequently overlooked [10].

Previous studies aimed to investigate the impact of autonomic dysfunction on functional outcomes in patients with acute stroke. According to a study by Xiong et al., patients with significant autonomic dysfunction, as evaluated using the Ewing classification, demonstrate poor functional outcomes following acute stroke [11,12]. Autonomic dysfunction resulting from stroke can also impact the cardiovascular system, demonstrating reduced heart rate variability, among other factors. Bassi et al. suggested that evaluating heart rate variability before initiating rehabilitation in patients with subacute stroke can improve the chances of achieving satisfactory functional recovery. Additionally, the study suggested that inappropriate cardiovascular responses to rehabilitation therapy can diminish the capacity for rehabilitation and contribute to poor functional outcomes [13]. Other studies have highlighted the importance of autonomic dysfunction. In a prospective observational cohort study, autonomic dysfunction was reported as highly prevalent in patients with mild acute ischemic stroke and persisted for 6 months [5]. Therefore, autonomic dysfunction is a major concern in patients with stroke.

This study aimed to investigate the effects of autonomic dysfunction on functional outcomes in patients with acute stroke. We hypothesized that patients with severe autonomic dysfunction would have poor functional outcomes within 6 months. 

## 2. Materials and Methods

### 2.1. Subjects and Study Design

This retrospective study was conducted at a single center and included 22 patients (15 males and 7 females) with stroke onset from March 2018 to February 2023. The inclusion criteria were as follows: (1) individuals with initial stroke diagnosed by computed tomography or magnetic resonance imaging interpreted by a specialized radiologist, (2) those who underwent an autonomic function test about 6 months after stroke onset, (3) those followed up with a functional outcome test about 6 months, and (4) those aged 18 years or older. The exclusion criteria were as follows: (1) medical history of other pulmonary or cardiogenic diseases and (2) other disorders that could cause autonomic dysfunction, such as diabetes mellitus, neurodegenerative disorders, autoimmune diseases, and cancer. This study was approved by the Institutional Review Board approval (IRB No. 05-2023-082).

### 2.2. Autonomic Function Test

Autonomic functions, including that of the sympathetic, parasympathetic, and sudomotor systems, were measured [11]. Sympathetic and parasympathetic nervous systems were evaluated using a Finometer Pro (Finapres Medical Systems, Amsterdam, The Netherlands). Sudomotor function was assessed using a Q-SweatTM Quantitative Sweat Measurement System (WR Medical Electronics Co., Maplewood, MN, USA) (Figure 1). The assessment was conducted by a medical laboratory technologist with over 5 years of experience. The CASS, which was developed by Low at the Mayo Clinic in 1993, was used as a tool for evaluating autonomic dysfunction [14]. The CASS evaluates autonomic function by assessing cardiovagal, adrenergic, and sudomotor function, with scores ranging from 0 to 3 for cardiovagal and sudomotor function and 0 to 4 for adrenergic function, resulting in a total score of 0 to 10. Each item was evaluated using the method proposed by Low et al. A score of 1–3 was considered mild, 4–6 as moderate, and 7–10 as severe. The severity of autonomic dysfunction in patients with acute stroke was classified into mild, moderate, and severe groups, and their initial functional outcomes were compared with the follow-up results [14]. The CASS had a sensitivity of 94% and a specificity of 100% for detecting severe autonomic dysfunction (Table 1) [14].

#### 2.2.1. Sympathetic Test (Adrenergic Index)

##### Blood Pressure Response to Postural Change

After wearing an arm cuff and resting in the supine position for 5 min, systolic blood pressure (SBP) and diastolic blood pressure (DBP) were measured. The tilt table was then inclined to 70°. SBP and DBP were measured again after standing for 1, 3, and 5 min. Orthostatic hypotension was considered if there was a decrease of ≥30 mmHg in SBP or a decrease of ≥10 mmHg in DBP, regardless of symptoms. During the examination, a minimum of two medical laboratory technologists closely monitored for neurological symptoms, such as dizziness, disorientation, and headaches.

##### Beat-to-Beat Blood Pressure

After a 5 min resting period, an arm cuff was applied to the brachial artery, and beat-to-beat blood pressure was continuously monitored using a screening device. Blood pressure, including systolic, diastolic, and mean blood pressure, and heart rate were measured serially. The patients were instructed to exhale forcefully for 15 s, and the decline in blood pressure during phase II was measured. The adrenergic index was classified as 0 points for normal; 1 point for a mean blood pressure decrease of >20 mmHg but <40 mmHg in early phase II without returning to baseline in late phase II or phase IV; 2 points for a mean blood pressure decrease of >20 mmHg but <40 mmHg in early phase II with no late phase II or phase IV; 3 points for a mean blood pressure decrease of >40 mmHg in early phase II with absent late phase II and phase IV; and 4 points for criterion 3 plus orthostatic hypotension, defined as a decrease in systolic blood pressure > 30 mmHg or diastolic blood pressure > 10 mmHg.

#### 2.2.2. Parasympathetic Test (Cardiovagal Index)

##### Heart Rate Response to Deep Breathing

The patient was equipped with a heart rate monitoring device for electrocardiography (ECG) and performed deep breathing exercises. The patient rested for 5 min before the test. The heart rate response was evaluated by performing 6 breaths with deep and steady inhalation and exhalation for 5 s, each in the supine position. Before the test, they practiced adequate breathing time and intensity under the supervision of a physiotherapist who provided verbal cues. The results were obtained as the ratio of the longest R-R interval to the shortest R-R interval.

##### Heart Rate Response to Valsalva Maneuver

The participants exhaled forcefully for 15 s while maintaining an expiratory pressure of 40 mmHg, measured with a mouthpiece connected to a pressure transducer while monitoring the ECG. After exhalation, the subjects performed tidal breathing for a relaxation period of 45 s. Prior to the test, a rehearsal was conducted, allowing forceful exhalation for 15 s after a 5 min rest period. The Valsalva ratio is the change in heart rate during the Valsalva maneuver. The test was performed at least thrice over a period of 1 min. The ratio was calculated by dividing the maximum R-R interval recorded during the test by the minimum R-R interval. Evaluation of the cardiovagal index was defined as 0 points for normal values, 1 point for a heart rate decrease of <50% of the lower limit of normal during deep breathing or the Valsalva maneuver, 2 points for a heart rate decrease of 50% during deep breathing or the Valsalva maneuver, and 3 points for a heart rate decrease of 50% during both deep breathing and the Valsalva maneuver.

#### 2.2.3. Sudomotor Test (Sudomotor Index)

##### Quantitative Sudomotor Axon Reflex Test

Sudomotor function was measured by assessing the volume and latency of sweat in both arms and proximal legs. Iontophoresis boxes were attached to the volar side of the forearm and lateral aspect of the proximal lower legs of patients in a supine position at rest. After a 2 min stimulation to induce acetylcholine release, sweat latency and volume were measured for 5 min. According to the CASS, 0 points were considered normal; 1 point indicated abnormal sweat volume at one site with a value of ≥50% of the lower limit; 2 points indicated a value of <50% of the lower limit at one site; and 3 points indicated values of <50% of the lower limit at two or more sites.

### 2.3. Functional Outcome

A medical laboratory technologist with over 5 years of experience evaluated the participants’ functional ability both initially and again after approximately 6 months since functional recovery mostly occurs within 6 months of stroke onset [15,16,17]. The Modified Barthel Index (MBI), Berg Balance Scale (BBS), Functional Ambulatory Category (FAC), and modified Rankin Scale (mRS) were used to compare functional outcomes in this study. The MBI measures performance in ADL, comprises 10 items, and is scored out of 100 points, with higher scores indicating higher functional status. The BBS is an objective tool used to assess dynamic balance. It comprises 14 items, each rated on a scale of 0–4 points, resulting in a total score of 56 points. The FAC classifies walking ability into six levels based on the degree of physical support, evaluated according to the criteria used in the study by Mehrholz et al. [18] The mRS is a scale reflecting global disability grading from 0 to 6. Grades 0 and 6 indicate no symptoms at all and death, respectively, as evaluated according to the criteria used in the study by Banks et al. [19].

### 2.4. Statistical Analysis

Statistical analysis was performed using SPSS software (version 26.0; IBM Corp., Armonk, NY, USA). Data normality was tested using the Kolmogorov–Smirnov and Shapiro–Wilk tests. Demographic data were analyzed using the chi-square test. One-way analysis of covariance (ANCOVA) was used to compare differences in functional outcome values between groups according to the severity of autonomic dysfunction for the initial and follow-up periods. Scores are presented as mean ± standard deviation. Statistical significance was set at *p*-value < 0.05. 

## 3. Results

A total of 156 patients with stroke were screened, and 22 met the inclusion criteria (Figure 2). Among 156 patients, 12 individuals with pulmonary disease and 6 with cardiovascular disease were initially excluded. During the selection process of 138 patients, those with diabetes mellitus, neurodegenerative disease, autoimmune disease, and cancer were excluded. Instances where an individual had two or more coexisting conditions were documented. In particular, 102 individuals had diabetes mellitus, 6 had neurodegenerative disease, 3 had autoimmune disease, and 18 had cancer, resulting in a total exclusion of 116 individuals. Out of the 156 assessed for eligibility, only 22 satisfied the inclusion criteria. The mean age of patients with acute stroke was 59.72 ± 16.68, and among the 22 patients, 15 were male. Stroke risk factors included hypertension (n = 9, 40.9%) and dyslipidemia (n = 7, 31.8%). Stroke types included infarction and hemorrhage in 12 and 10 patients, respectively. Stroke was unilateral in 16 patients (eight on the left, eight on the right) and bilateral in six patients. The patient demographic characteristics are presented in Table 2. Each group was classified as mild (n = 9), moderate (n = 6), or severe (n = 7) based on CASS. The days of follow-up represent the duration elapsed from the initial evaluation to the subsequent assessment of functional outcomes. In the mild group, this period was 164.0 ± 19.8 days; in the moderate group, it was 171.3 ± 7.7 days; and in the severe group, it was 179.3 ± 15.5 days. Similarly, the days of the autonomic function test denote the number of days after stroke onset when the autonomic function test was conducted, with durations of 34.0 ± 22.0 days in the mild group, 54.8 ± 33.3 days in the moderate group, and 59.3 ± 43.4 days in the severe group. Functional outcomes were summarized as the combined scores from both the initial and follow-up evaluations. No statistically significant differences in demographic factors were observed among the groups based on the provided data.

### Comparison of Functional Outcomes between Mild, Moderate and Severe Autonomic Dysfunctions

The changes in functional outcomes were compared among the three groups classified according to CASS (Table 3). The values in Table 3 represent the mean ± standard deviation of the functional outcomes at follow-up, as analyzed using ANCOVA. All four functional outcomes showed statistically significant differences between the three groups. After post hoc analysis using Tukey comparison, the mild group exhibited a statistically significant difference from the moderate and severe groups. However, no significant difference was observed between the moderate and severe groups.

## 4. Discussion

According to the findings, differences in the initial functional outcomes and recovery after follow-up were shown among the three groups. Significant differences were observed in all four functional outcomes, including MBI, BBS, FAC, and mRS. This result suggests that the severity of autonomic dysfunction influences the functional prognosis in patients with acute stroke. Additionally, when comparing the differences between the two groups, the mild group showed statistically significant differences compared to both the moderate and severe groups in post hoc analysis. This indicates that functional recovery was more favorable in the mild group, as classified by the CASS. However, there was no significant difference between the moderate and severe groups, suggesting that autonomic dysfunction above the moderate level was associated with poorer functional recovery than mild autonomic dysfunction.

Autonomic dysfunction can lead to impaired functional outcomes for several reasons. First, symptoms due to autonomic dysfunction, including dizziness and orthostatic hypotension, can reduce the frequency and intensity of rehabilitation. Second, the dysregulation of the sympathetic and parasympathetic nervous systems can lead to insufficient blood supply to the injured brain tissue. Third, the autonomic nervous system is important for regulating stress and maintaining homeostasis in response to the brain’s perception of stressors such as acute and chronic stroke [20]. Fourth, autonomic dysfunction after acute stroke can lead to downregulation of cardiac function. According to a study by Scheitz et al., this phenomenon is called stroke–heart syndrome and is characterized by changes in cardiomyocyte metabolism, the dysregulation of the leukocyte population, and vascular changes, which can lead to stroke-induced cardiac stress and a 2- to 3-fold increased risk of short-term mortality [21]. Fifth, the overactivity of the sympathetic nervous system and increased levels of norepinephrine can cause stroke-induced immune suppression (SIS), predisposing patients to infections. Furthermore, the occurrence rate of pneumonia and urinary tract infections has been reported to be up to 30% following acute stroke [22]. Sympathetic overactivity enhances the production of pro-inflammatory cytokines and decreases lymphocyte activity [23]. According to a study by Ruhnau et al., functional deficits were reported in natural killer cells and T lymphocytes as well as in monocytes and granulocytes [24]. Furthermore, the occurrence rate of pneumonia and urinary tract infections has been reported to be up to 30% following acute stroke [22]. Infections contribute to higher mortality rates and poorer functional outcomes. Considering the association of SIS with functional recovery, further vigilant investigation is needed in future studies. Autonomic dysfunction after acute stroke is a risk factor that should not be overlooked as it can also affect functional recovery and mortality.

The significance of autonomic dysfunction has recently garnered attention in various studies. It is becoming increasingly important not only in the context of stroke but also in predicting and analyzing autonomic dysfunction in various other conditions. In a study targeting patients with Alzheimer’s disease, a model was developed for the non-invasive assessment of the autonomic nervous system using heart rate variability to evaluate the pathophysiology and progression of the disease [25]. According to Konstantin et al., short-term measurements of the root mean square of successive differences in patients with Parkinson’s disease can be utilized to assess parasympathetically impaired cardiac modulation [26]. Additionally, Guoliang et al. investigated the relationship between autonomic function using heart rate variability and cognitive performance in patients with cerebral small vessel disease [27]. Recent research suggests that autonomic dysfunction is gaining new prominence in diseases affecting the central nervous system. This study holds significance as evidence confirming the importance of autonomic dysfunction in patients with acute stroke.

This study has some limitations. First, the generalizability of the study findings is limited due to the single-center design. Second, this study is limited by the small sample size and subsequent low statistical power. The limited enrollment compared to screening patients for approximately 5 years is attributed to the exclusion of individuals with diabetes mellitus. Diabetes mellitus is a risk factor for stroke; therefore, it is likely that patients with stroke had this disease as a common comorbid condition. Future studies with larger sample sizes and multiple centers are required to confirm these findings. Third, it is difficult to attribute changes in functional outcomes solely to autonomic dysfunction, as other factors such as the different environments, including the intensity and time of treatments received by each patient, could have contributed to the results. Fourth, as this was a retrospective study, it was not possible to control for all the factors that may have influenced the autonomic function tests. For example, the types of food consumed by the patient before the test, the duration of fasting, resting time, level of physical activity, and room temperature. Future studies should control for various factors to obtain more reliable results. Finally, this study did not differentiate between ischemic and hemorrhagic stroke, which have different pathologies and severities. Separating these two types of strokes may provide more specific information regarding the relationship between autonomic dysfunction and stroke prognosis.

## 5. Conclusions

In conclusion, this study provides relevant evidence of the association between autonomic dysfunction and functional outcomes in patients with acute stroke. We highlight the importance of early screening for autonomic dysfunction in patients with acute stroke to predict prognosis and the need to regulate autonomic dysfunction.

## Figures and Tables

**Figure 1 brainsci-13-01694-f001:**
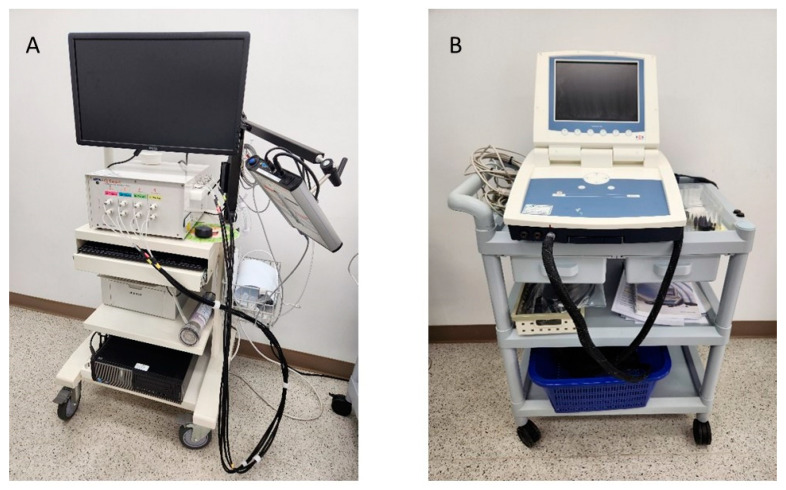
Autonomic function test devices. (**A**) Finometer Pro measured the sympathetic and parasympathetic function. (**B**) Q-Sweat™ Quantitative Sweat Measurement System measured the sudomotor function.

**Figure 2 brainsci-13-01694-f002:**
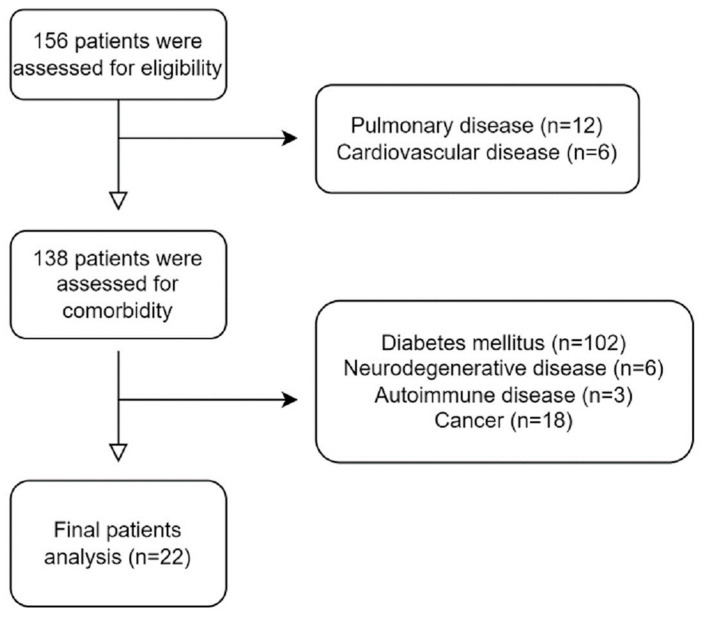
Flow chart of patient enrollment.

**Table 1 brainsci-13-01694-t001:** Composite autonomic scoring scale (CASS).

	Category		Score
Evaluation items	Cardiovagal	Heart rate on deep breathing and Valsalva maneuver	0–3
	Adrenergic	Phase II, IV during Valsalva maneuver (Orthostatic hypotension)	0–4
	Sudomotor	Q-SART	0–4
Interpretation		Mild	1–3
		Moderate	4–6
		Severe	7–10

Q-SART, quantitative sudomotor axon reflex test.

**Table 2 brainsci-13-01694-t002:** Characteristics of patients with acute stroke according to classification of autonomic dysfunction.

	Mild Group (n = 9)		Moderate Group (n = 6)		Severe Group (n = 7)	*p* Value
Age	59.9 ± 17.1		60.3 ± 18.2		57.2 ± 17.5	0.077
Sex (male), n (%)	6 (66.7)		4 (66.7)		5 (71.4)	0.975
BMI	24.1 ± 4.1		24.5 ± 4.4		24.4 ± 4.6	0.402
Hypertension, n (%)	2 (22.2)		3 (50.0)		4 (57.1)	0.322
Dyslipidemia, n (%)	4 (44.4)		1 (16.7)		2 (28.6)	0.514
Stroke type, n (%)						
Infarction	8 (88.9)		2 (33.3)		2 (28.6)	0.026
Hemorrhage	1 (11.1)		4 (66.7)		5 (71.4)	
Stroke side, n (%)						
Right	2 (22.2)		4 (66.7)		2 (28.6)	0.052
Left	6 (66.7)		1 (16.7)		1 (14.3)	
Both	1 (11.1)		1 (16.7)		4 (57.1)	
Days of follow-up (d)	164.0 ± 19.8		171.3 ± 7.7		179.3 ± 15.5	0.239
Days of autonomic function test (d)	34.0 ± 22.0		54.8 ± 33.3		59.3 ± 43.4	0.676
Functional outcomes	Initial	After	Initial	After	Initial	After
MBI	35.3 ± 26.7	80.1 ± 18.5	13.8 ± 20.4	27.7 ± 31.1	7.6 ± 17.5	12.1 ± 21.7
BBS	22.3 ± 18.4	40.0 ± 13.5	3.0 ± 3.5	9.0 ± 11.6	5.3 ± 14.0	7.7 ± 19.1
FAC	1.2 ± 1.2	3.8 ± 1.3	0.2 ± 0.4	1.2 ± 1.3	0.3 ± 0.8	0.3 ± 0.8
mRS	3.7 ± 1.0	1.7 ± 0.9	4.7 ± 0.5	4.0 ± 1.3	4.9 ± 0.4	4.7 ± 0.5

MBI, modified Barthel index; BBS, Berg balance scale; FAC, functional ambulatory category; mRS, modified Rankin scale. Values are presented as the mean ± standard deviation. The chi-square test was used to compare inter-group variances.

**Table 3 brainsci-13-01694-t003:** Comparison of functional outcomes between the three groups classified according to CASS.

Functional Outcomes	MBI	BBS	FAC	mRS
**Mild**	69.17 ± 6.14 ^a^	31.08 ± 3.49 ^a^	3.49 ± 0.40 ^a^	2.03 ± 0.32 ^a^
**Moderate**	32.73 ± 6.96 ^b^	16.21 ± 4.02 ^b^	1.40 ± 0.47 ^b^	3.81 ± 0.35 ^b^
**Severe**	21.87 ± 6.73 ^b^	13.01 ± 3.65 ^b^	0.46 ± 0.43 ^b^	4.41 ± 0.34 ^b^
**F**	12.93	6.10	12.40	11.08
***p* value**	<0.001 *	0.009 *	<0.001 *	0.001 *

MBI, modified Barthel index; BBS, Berg balance scale; FAC, functional ambulatory category; mRS, modified Rankin scale; comparison of functional outcomes between the three groups; one-way ANCOVA was used. Values are presented as the mean ± standard deviation. * *p* < 0.05. ^ab^ post hoc analysis using Tukey comparison.

## Data Availability

The data presented in this study are available on request from the corresponding author. The data are not publicly available due to their containing information that could compromise the privacy of research participants.

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
