# Peer review of "The Effects of Autonomic Dysfunction on Functional Outcomes in Patients with Acute Stroke"

_brainsci, 2023, doi:10.3390/brainsci13121694_

Round 1
Reviewer 1 Report
Comments and Suggestions for Authors
First of all, I want to thank you for the possibility of reviewing this work. I find the study very interesting and very novel.
I find its structure clear and easy to read.
Just a comment: it strikes me that of 156 stroke patients, only 22 met the inclusion criteria, because this tells us that many patients will not be able to benefit from this work. Above all, I want to highlight the fact that patients with diabetes are excluded, because diabetes is a very important risk factor for suffering a stroke, and patients who have suffered a stroke usually present a very high % of diabetes as a concomitant disease.
I don't know if it is possible to do something about this section.
Author Response
Thank you for your detailed and accurate review. I have attached my response in a Word file.

Reviewer 2 Report
Comments and Suggestions for Authors
In first it was a great job and good idea not new but did well
i think you not need all these number of data especially if the results is expected
need to concise the data extraction and why using small size in this long periods I think I was the main weakness
discussion need more editing and references need also to improve
Comments on the Quality of English Language
Red to improve
Author Response

(The authors gave the same response as above.)

Reviewer 3 Report
Comments and Suggestions for Authors Cha et al are presenting an exploratory study addressing the impact of post-stroke dysautonomia on the outcomes. The study is of interest but there are several points to be addressed : 1." Stroke can impact not only the central but also the peripheral nervous system, such as the autonomic nervous system." This is a misconception. Stroke causes damage of the central autonomic network which causes peripheral manifestations. 2. "Autonomic innervation injury affecting the gastrointestinal system 58 often results in gastroparesis, which manifests as constipation, vomiting, and abdominal 59 distension. Additionally," - This is not correct. There is no innervation injury. Replace by autonomic dysfunction affecting the gastrointestinal system... 3. "This study 81 highlights the importance of early screening for autonomic dysfunction in patients with 82 acute stroke to predict functional prognosis, and the need to regulate autonomic dysfunc- 83 tion." - This paragraph sounds like a conclusion. Remove it from the introduction. 4. In the methods section : exclude amyotrophic lateral sclerosis from the list of diseases with autonomic dysfunction (it is a marginal manifestation) 5. "A score of 1–3 was considered mild, 4–6 as moderate, and 7–10 as severe. The severity 111 of autonomic dysfunction in patients with acute stroke was classified into mild, moderate, 112 and severe groups and their initial functional outcomes were compared with the follow- 113 up results. " - provide the appropriate rationale or supportive literature for this division 6. Provide a figure with flowchart of patient inclusion and exclusion in the study 7. Remove references to table in the discussion section 8. In the discussion : "This result indicates that the severity of autonomic dysfunction influences the function" - replace indicates by suggests in this sentence and in the rest of discussion section 9. "Second, dysregulation of the sympathetic and parasympathetic nervous systems can lead to insufficient blood supply to the injured brain tissue, which can interfere with penumbra recovery" - After 48 h , stroke lesion is well established , therefore autonomic dysfunction to justify outcomes based on the effect on penumbra is not appropriate in the context of this study. 10. Line 260 - d granulocytes 24. -the reference is not formatted Comments on the Quality of English LanguageImprovement is needed.
Author Response

(The authors gave the same response as above.)

Round 2
Reviewer 3 Report
Comments and Suggestions for Authors
The authors have responded to all my comments.